# Anti-Thermal Shock Binding of Liquid-State Food Waste to Non-Wood Pellets

**Bruno Rafael de Almeida Moreira** [1],*[ID], **Ronaldo da Silva Viana** [2][ID], **Victor Hugo Cruz** [2],
**Paulo Renato Matos Lopes** [2], **Celso Tadao Miasaki** [2], **Anderson Chagas Magalhães** [2],
**Paulo Alexandre Monteiro de Figueiredo** [2], **Lucas Aparecido Manzani Lisboa** [2],
**Sérgio Bispo Ramos** [2], **André May** [3] **and José Claudio Caraschi** [4]

[1] Department of Phytosanitary, Rural Engineering and Soils, School of Engineering, São Paulo State University (Unesp), Ilha Solteira, São Paulo 15385-000, Brazil

[2] Department of Plant Production, College of Agricultural and Technological Sciences, São Paulo State University (Unesp), Dracena, São Paulo 19900-000, Brazil; ronaldo.viana@unesp.br (R.d.S.V.); cruz.v.h@outlook.com (V.H.C.); prm.lopes@unesp.br (P.R.M.L.); celso.t.miasaki@unesp.br (C.T.M.); ac.magalhaes@unesp.br (A.C.M.); paulo.figueiredo@unesp.br (P.A.M.d.F.); lucas.lisboa@unesp.br (L.A.M.L.); sergio.bispo@unesp.br (S.B.R.)

[3] Brazilian Agricultural Research Corporation (Embrapa), Jaguariúna, São Paulo 13820-000, Brazil; andre.may@embrapa.br

[4] São Paulo State University (Unesp), Itapeva, São Paulo 18400-000, Brazil; j.caraschi@unesp.br

* Correspondence: bruno_rafael.m05@hotmail.com; Tel.: +55-18-3821-7476

**Abstract:** The development and implementation of strategies to assist safe and effective transport and storage of pellets in containers and indoor facilities without heating systems are challenging. This study primarily aimed to reshape the organic fraction of municipal solid waste into a liquid-state binder in order to develop freezing–defrosting-proof non-wood pellets. The introduction of the standard solution of food waste into the process of pelleting consisted of stirring it together with the residual biomass from distillation of cellulosic bioethanol or alternatively spraying very fine droplets on the layer of the starting material before it entered the pilot-scale automatic machine at 200 MPa and 125 °C. The addition by spraying of carbohydrate-rich supplement boiled for five minutes caused the pellets to show increases in apparent density (1250.8500 kg·m$^{-3}$), durability (99.7665%), and hydrophobicity (93.9785%), and consistently prevented them from suffering severe mechanical fracture by thermal shock. The fractal dimension of breakpoints, cracks, and delamination on the finished surface for these products was the smallest at 1.7500–1.7505. Sprayed pellets would fall into the strictest grid of products for residential heat-and-power units, even after freezing and defrosting. The conclusion is therefore that spraying can spectacularly ensure the reliability of liquid-state food waste as an anti-thermal shock binder for non-wood pellets.

**Keywords:** agro-industrial residue; defrosting; durability; fractal analysis; fractures; freezing; storage; municipal solid waste; transportation; waste-to-energy

## 1. Introduction

The macroeconomic clusters of Africa, East Asia and Pacific, Latin America and the Caribbean, the Middle East and Africa, and the Nations of Organization for Economic Co-operation and Development (OECD) together generated 3.5 billion tons municipal solid waste (MSW) per day in 2017. The global production of MSW is likely to reach 6 billion tons per day by 2025, equivalent to 9.75 kg per capita per day [1]. Disposal of MSW in landfills or simply off-site is not responsible, as it is expensive, detrimental to the environment, and wasteful [2–5]. Composting, anaerobic digestion,

and fermentation, as well as incineration or combustion, gasification, torrefaction, hydrothermal carbonization, steam explosion, pyrolysis, and liquefaction are the simplest and wisest waste-to-product (WtP) and waste-to-energy (WtE) pathways to convert the organic fraction of MSW to bio-oil, syngas, and solid biofuels such as biochar and pellets [6–9]. The technical use of food waste as an ingredient (whether as a base or a supplement) in the compaction of biomass for the production of fuel-grade biosolids is not consistent. Further investigations on preprocessing techniques to make its re-use through the path of pelleting suitable are, therefore, necessary for technical reasons.

Pellets are solid biofuels from the processing of woody and/or non-woody biomass through the versatile path of pelleting [10–12]. These fuel-grade biosolids are replete with commendable environmental, economic, and social benefits for the increasing world population, including easier, more efficient, and cheaper transportation and storage of biomass; mitigation of emissions of greenhouse gases into the atmosphere; and improvement of conditions of people living in rural zones where accessibility to the national electricity grid is difficult [13–19]. The process of pelleting usually comprises the major stages of pre-production, production, and post-production. The tasks of pre-production or preprocessing involve gathering, characterizing, and conditioning the starting material by drying, grinding, steaming, and blending or adding lubricant, plasticizer, or binder if further improvements are required for it to become grindable, flowable, compressible, and compactable before entering the machine [10].

Binders work like adhesive resins for making polymers and fiberboard [20]. The strictest international specifications set the concentration of additive in the mixture as no greater than 2% of the total mass. Otherwise, the process of pelleting can become uncontrolled, causing wear to the machine and reducing the productivity, quality, and marketability of the pellet as a consequence of unacceptable but controllable failure. Starch, cereal flour, lignin, coffee meal, bark, inedible vegetable oils, molasses, and water-soluble carbohydrates are the most affordable and most reliable solid and liquid binding agents for manufacturing pellets at a large scale [21–23]. Starchy and sugary binders are greatly effective for saving energy consumption during compaction, and for enhancing the capacity of the pellet to firmly resist deformation by abrasive forces like thermal shock which can easily promote fractures in the form of breakpoints, cracks, splashing, and delamination [24]. However, they could be more suitable for furnishing multi-purpose feedstocks for the food, drug, personal care, and cosmetic industries. The development and implementation of competing additives to prevent the continuously evolving global market of fuel pellets from facing potential conflicts of interest for mutual feedstock and to avoid crises are, therefore, necessary for both ethical and economic reasons. The organic fraction of MSW could be an option to assist in resolving this challenge easily and cheaply.

Several thermal-chemical preprocessing techniques to enhance the performance of organic and inorganic additives in forming homogeneous, consistent pellets exist. The most popular is steam explosion [25–27]. Stirring and mixing are the most traditional methods of introducing additives into pelleting. Spraying is gradually emerging as an alternative, and therefore requires further investigations with respect to its suitability for applying organic binders through droplets [28].

The global production of wood pellets increased impressively from 5 Mt to upwards of 30 Mt by 2017. In the year 2017, consumption of wood pellets in bags for heating in 500-kW boilers by residential customers in Latvia, Germany, France, and many other European countries reached 9 Mt, at the price of €250 per metric ton. The European Union (EU) is, of course, the world's largest producer and consumer of wood pellets for heating and power [29–31]. Meanwhile, North Korea and Japan account for the majority of the steadily evolving Asian markets, thanks to governmental policies and programs like Trade-in Tarif and Renewable Portfolio Standard. The global market of pellets is growing quickly, and is likely to reach 60 Mt by 2025 [32,33].

The main intercontinental trade of pellets occurs between America and Europe. In 2018, the United States and Canada traded approximately 7.5 billion metric tons with the European Union—the United Kingdom, Denmark, and Italy accounted for the EU's largest importers [34]. This journey may take few weeks or even couple of months, depending on origin, destination, kind and effectiveness of

transport, and terminal time plans. North America's transportation of pellets in ship containers, from Canada and the Eastern United States to Northern Europe, usually takes no more than 45 days [35].

Safe and effective of transportation and storage are of great importance for the utilization of pellets as solid biofuels for heating and power in biomass-burning household stoves, gasifiers, and furnace-boiler systems in restaurants and coal-firing stations [36]. The pellet's integrity is key to minimizing potential risks and ensuring comfort for the manufacturer and consumer, as well as for feeding thermal conversion equipment [37,38]. In the winter, unpredictable fluctuations in temperature and relative humidity of the air can challenge the transport and storage of pellets for long periods in containers and indoor facilities with no heating systems such as seasonal silos and warehouses [39,40].

The process of freezing and subsequent defrosting can reduce pellet density, durability, hydrophobicity, and thermal power without difficulty. If the pellet is mechanically and thermally inconsistent, it can no longer successfully perform as an effective solid biofuel [41–43]. Additionally, sudden changes in temperature and relative humidity of the air can cause the pellet to release dust and fines easily to the environment and generate greenhouse gases such as CO, $CO_2$, and $CH_4$, as well as light volatile biocompounds [44–46]. Dust and fines are hazardous for fires in indoor facilities and are dangerous to human health, as these particulates are potential causes of occupational diseases in workers dealing with handling, transportation, and storage of poorly durable material, and can affect the final consumer. Storage for over 20 months can reduce the durability of the pellet by 30%, thus causing it to spontaneously generate substantial quantities of dust and fines if the facility is microclimatically inadequate for long-term storage [47–50].

Evidently, wood pellets are at the center of both production and consumption of this strict category of solid biofuels worldwide. However, residues of softwood and hardwood themselves will be not able to fulfill the future demand for wood pellets. Agricultural, agro-industrial, and municipal residues are potential sources of suitable feedstock for the development of pellets that do not involve woody biomass. Stoves, gasifiers, furnaces, and boilers can burn pellets from by-products of cereal crops and grasses without any difficulties [43]. The purpose of this study is to produce non-wood pellets from pressing residual biomass from distillation of cellulosic bioethanol with the addition of an organic fraction of municipal solid waste.

Cellulosic bioethanol complies with the standards for the category of second-generation biofuels for transportation [51,52]. It will be an eventual replacement for gasoline to power the multi-stroke Otto engine [53–55]. Cellulosic bioethanol, or simply 2G bioethanol, can be obtained from the industrial processing of lignocellulose through the major stages of pre-treatment, enzymatic hydrolysis, fermentation, and separation/distillation [56–60]. The final stage of the process separates the bioethanol from the broth through molecular sieves or membranes, and then concentrates the product to the optimal fuel grade by distillation. In the downstream steps, burning of residual biomass from distillation can power upstream steps with heat and steam. However, this solid effluent is often heterogeneous in size and shape of particles, and has low density, is highly hygroscopic, and poorly energetic. These disadvantages make in loco management and storage as well as transportation of residual biomass to areas located away from the sugar-energy plant difficult and expensive [61–63]. The development and implementation of strategies targeting downstream steps to convert this challenging co-product to biofuels and fine chemicals to ensure second-generation bioethanol which is feasible to fabricate and competitive with (bio)fuels for transportation, like gasoline and first-generation bioethanol, are, therefore, necessary for both economic and technical reasons. The process of pelleting could be an option for simple and wise recycling.

Therefore, this timely study was aimed at testing the technical viability of recycling food waste into a liquid-state binder to produce freezing–defrosting-proof non-wood pellets from alternatively processing residual biomass from the separation/distillation of cellulosic bioethanol.

## 2. Materials and Methods

### 2.1. Pre-Production

### 2.1.1. Infrastructure

The infrastructure of the Laboratory of Machinery and Mechanics of the College of Agricultural and Technological Sciences, São Paulo State University (Unesp), Campus of Dracena, São Paulo, Brazil, was used. A pilot-scale set for pelleting comprised a feeder silo, an automatic pelletizer machine, and a vibratory screener with a cooler, all supplied from Lippel$^{TM}$ (Figure 1). Details of the set are in Table S1 (Supplementary Material).

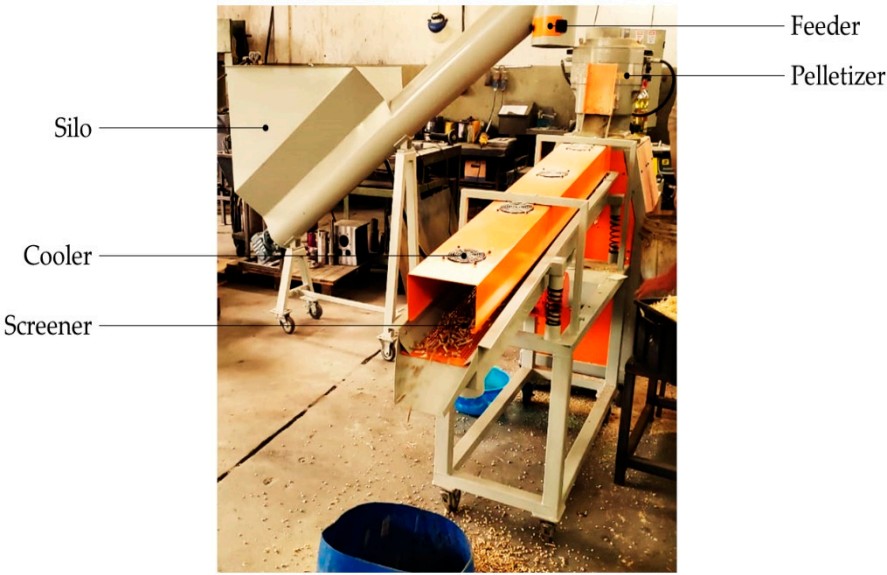

**Figure 1.** Pilot-scale set of pelleting.

### 2.1.2. Starting Material and Supplement

The starting material came from sugar-energy plants in Southeast Brazil. This was the residual biomass from the separation/distillation of cellulosic bioethanol from sugarcane bagasse. The supplement was from local restaurants. The food waste mainly consisted of cooked rice, spaghetti, and steak. The temperature and relative humidity of the air during the initial storage of materials in 5-kg hermetic polyethylene bags in the laboratory to prevent them from exposure to adverse weather agents which might negatively influence their performance upon briquetting and further analytical procedures were 22.5 ± 5 °C and 55 ± 4.5%, respectively, over three days.

Characterization

The physical–chemical characterization of the materials consisted of formally carrying out proximate and ultimate analyses, in triplicate, as per the norms of American Standard for Testing and Materials (ASTM) summarized in Table S2 (Supplementary Material).

### 2.2. Production

### 2.2.1. Experimental Planning

An experiment testing the technical viability of developing food waste as a liquid-state additive to non-wood pellets employed an orthogonal uniform design: $U_{21}(3^2 \times 2^1)$, factorial A × B × C, corresponding to time of boiling, method of addition, and storage condition. Each test comprised five replicates (Table 1).

**Table 1.** Orthogonal tests for the factorial experiment of developing food waste as a liquid-state addition to non-wood pellets from residual biomass from the distillation of cellulosic bioethanol.

| Level | Factor | | |
|---|---|---|---|
| | **A, Time of Boiling, min** | **B, Method of Addition** | **C, Storage Condition** |
| 1 | 0 | Stirring | Normal |
| 2 | 5 | Spraying | Freezing |
| 3 | 10 | | Defrosting |

### 2.2.2. Setup

#### Pre-Treatment of the Materials

The pretreatment of the materials consisted of conventional drying in a horizontal airflow drying oven at 65 °C until constant mass, milling in a Wiley knife mill, and then sieving in stainless-steel wire cloth with holes of 0.25 mm as per Lu et al. [64] and Mendoza Martinez et al. [65].

#### Preparation of Liquid-State Additive

The preparation by thermal-chemical preprocessing of the liquid-state additive consisted of the boiling (or not) of 0.5 kg food waste in powder in 2 L of deionized water at 100 °C for 5 and 10 min, and then filtering the solution through filter paper for the removal of solids in order to concentrate it into water-soluble carbohydrates finer than 0.005 mm, adapting the method of Zhang et al. [28].

#### Supplementation of the Starting Material

The traditional supplementation consisted of stirring 2 kg of starting material with 1 L standard-solution of additive at 2% in stirrer working at 25 rpm, clockwise, for 10 min. The alternative supplementation consisted of spraying 1 L of standard solution on a tiny layer of 2 kg starting material on the tabletop with a portable micro-sprayer (Sagyma SW-775, Burkard Scientific, Uxbridge, UK) pressured by a dual-action airbrush compressor (Sagyma ASW-18, Burkard Scientific, Uxbridge, UK) at 155 kPa to produce very fine droplets (30 μm) at 0.05 m above the target, adapting the method of Evandro et al. [66].

#### Pelleting

The pilot-scale manufacturing of non-wood pellets started with filling the silo with 2 kg of material with or without addition by stirring or spraying of liquid-state food waste. The feeder, working at 75 kg·h$^{-1}$, fed the pelletizer intermittently with the load of material, which then slipped through the holes of flat die underneath pressing rollers of the machine at 200 MPa and 125 °C during compaction for shaping into single pellets. The thresholds for the pressure and temperature of the process strictly obeyed the ranges of 115–300 MPa [67,68] and 75–150 °C [10] for making pellets, optimally. The machine automatically ejected the pellets from the channel-forming die onto the screener at 1 g to prevent them from undergoing fractures by excessive vibration. The cooler caused the heat to flash off from the mass of pellets, which was then packed into 2-kg polyethylene packs for further analytical procedures to technically assess the operational efficiency of pelleting in converting biomass to pellets and the mechanical quality of pellets stored at normal and freezing temperatures to simulate thermal shock.

### 2.3. Post-Production

#### 2.3.1. Trial of Storage

The trial of freezing–defrosting ran for no longer than five days, with the tests being kept on a working benchtop at the temperature and relative air humidity of 25 °C and 65% and in a laboratory freezer at −20 °C and 70%, respectively, according to the method of Gilvari et al. [34].

### 2.3.2. Technical Analysis

The assessment of both the process of pelleting and quality of the product consisted of measuring/determining the efficiency of conversion of biomass into biosolids and the physical–mechanical variables, diameter, length, apparent density, durability, fines, resistance to breaking down by water, hygroscopicity, and fractal dimension of fractures by thermal shock immediately after compaction, 48 h after freezing, and 24 h after defrosting at room temperature.

Efficiency of pelleting: This variable, expressed as a percentage, measured the effect of addition on the behavior of the starting material upon compaction through the ratio of mass of pellets directly ejected from the screener onto the container placed on the ground in front of the machine and the mass of unconverted material recovered from the waste discharging system, according to Avelar et al. [69].

Diameter and length (standard EN 16127): The assessment of **Ø** and **L**, both expressed in millimeters, consisted of measuring the pellet transversally and longitudinally, respectively, with a hardened stainless-steel electronic digital caliper (MrToolz, FBA-ip54, 0.01-mm resolution), according to the method of Lu et al. [64].

Apparent density (standard EN 1503): The $\rho$, expressed as kilogram per cubic meter, was the mass-to-volume ratio of 0.01 kg sample in immersion in 0.1 L water in glass test tube, adapting the method of Liu et al. [70].

Durability and fines (standard ISO 17831-1): The determination of $\delta$ and *F*, both expressed as percentages, consisted of shocking the 0.05 kg sample against itself and against the walls of the rotating chamber (0.2 m length × 0.2 m width × 0.10 m depth) working at 50 rpm, clockwise, for 10 min, sieving the material in stainless-steel wire cloth with holes of 3.15 mm, and then weighing it on an analytical digital scale of 0.0001 g resolution (Shimazu, model ATX-220, Barueri, SP, BR) to calculate these variables through the ratio of final and initial mass of the sample through Equations (1) and (2), respectively, adapting the methods of Jackson et al. [71] and Tumuluru et al. [72].

$$\delta\ (\%) = \left(\frac{m_f}{m_i}\right)100 \tag{1}$$

$$F\ (\%) = 100 - \delta \tag{2}$$

where $\delta$ is the durability; *F* is the content of fines, $m_i$ is the initial mass in gram, and $m_f$ is the final mass in grams.

Resistance to breaking down by water: The determination of $\eta$, expressed as minutes, consisted of timing a 0.01-kg sample in immersion in 0.25 L of water in a 0.5-L glass beaker breaking at room temperature, adapting the method of Law et al. [73].

Hygroscopicity: The determination of *W*, expressed as a percentage, consisted of storing a 0.01 kg sample in wet chamber at 25 °C and 90%, for 24 h, and then weighing it to calculate this variable through the ratio of final and initial mass using Equation (3) by adapting the method of Avelar et al. [69].

$$W\ (\%) = \left(\frac{m_f - m_i}{m_f}\right)100 \tag{3}$$

Fractal dimension of mechanical fractures by thermal shock: To determine the *Df* computationally, we took photographs on 25 pellets randomly selected for each test using the camera of a conventional smartphone (Motorola, model Moto G6 Play, São Paulo, SP, BR) at 0.05 m above the target to capture the structure of non-Euclidian geometric pattern against the contrasting background of the box, at the resolution of 720 × 1440 pixels.

### 2.4. Data Analysis

The statistical analysis of the data set ran with traditional and non-traditional mathematics. We formally performed a descriptive analysis to measure the overall performance of pellets using

means, and range analysis to adequately determine the pairs of factors to accurately fit the interactive effect of time of boiling, method of addition, and storage condition to the physical–mechanical quality through the 2D contour plotting approach. To determine the fractal dimension of fractures by thermal shock we carried out the box-counting method. This is one of the simplest and most accurate approaches for applying fractal analysis to patterns defying understanding with classic Euclidean geometry [74]. The box-counting method determines the fractal dimension of the pattern by covering it with grids and then counting how many boxes are touching part of it in the 1D topological dimension. Graphically, the fractal dimension of the object is the slope of the curve when we plot the value of N(s) on the *y*-axis against the value of s on the *x*-axis. This time, N(s) is the number of grids covering the pattern, and the s is the size of box, or the inverse of magnification. The process advances iteratively with finer grids of smaller boxes. By shrinking the size of box, we more accurately capture the structure of the pattern with similar copies of itself at sizes and scales in the 2D immersive dimension. Therefore, the $D_f$ must range from 1 to 2 in power. The greater the slope of the curve, the more fractal the pattern, as it gains in complexity. On the contrary, the flatter the curve, the less fractal the pattern, as the richness of details does not increase with decreasing size of the box [75]. We performed the curation of the set of images through the non-binary equalization of frequency of RGB channels prior to running the box-counting method. In both the cases of 2D contour plotting and fractal analysis we implemented fuzzy logic to turn eventual ambiguities off to improve the accuracy of prediction and visualization of non-Boolean patterns [76]. The software used was R-project [77], which complies with and runs in several platforms. This multi-paradigm programming language provides a user-friendly environment for statistical computing and graphics.

## 3. Results

### 3.1. Proximal and Elemental Properties of the Materials

The proximal contents of fixed-carbon and ash were higher in the starting material than in the supplement, which was higher in water content. The largest portion of both the ingredients consisted of volatile matter (Table S3, Supplementary Material). The elemental contents of carbon and oxygen of the residual biomass were higher than those of the load of food waste, which was higher in contents of hydrogen and nitrogen.

### 3.2. Relative Performance of Addition by Spraying of Liquid-State Food Waste to Produce Freezing–Defrosting-Proof Non-Wood Pellets

The exact diameters and lengths were 6.0500, 6.0355, and 6.4050 mm, and 30.6050, 30.5375, and 30.9985 mm for the normal, frozen, and defrosted pellets exclusively consisting of residual biomass, respectively (Table 2). Therefore, expansion by defrosting transversally and longitudinally in pure products was 5.5425% and 1.2695%, respectively. Evidently, referential pellets themselves expanded more transversally than longitudinally after gradual defrosting at room temperature. Irrespective of the time of boiling, the diameters and lengths in pellets with addition by spraying were in the ranges of 6.0340–6.0410 and 29.5900–39.7200 mm immediately after pelleting, 5.9330–6.0015 and 29.5480–39.4500 mm after freezing, and 6.0500–6.0765 and 29.6830–39.8330 mm after defrosting, respectively. Thus, transversal and longitudinal expansion values in pellets with addition by spraying were 0.1490–0.6965% and 0.1820–0.3140%, respectively. These products with traditional addition by stirring were more unstable as they expanded more transversally and longitudinally, at 0.4745–1.5250% and 0.1135–0.4510%, respectively. Practically, spraying technically was superior to stirring in adding liquid-state food waste to develop pellets structurally stiffer to contraction by freezing and subsequent expansion by defrosting, while preserving much of the original geometric length-to-diameter ratio after thermal shock.

**Table 2.** Descriptive analysis for the diameters and lengths of non-wood pellets with addition by stirring or spraying of liquid-state food waste subjected to the process of freezing–defrosting.

| Time | Method | Storage Condition | | | Fluctuation | | |
|---|---|---|---|---|---|---|---|
| | | Normal, N | Freezing, F | Defrosting, D | N–F | F–D | N–D |
| | | Diameter, mm | | | Δ, % | | |
| Control | | 6.0500 ± 0.1200 | 6.0355 ± 0.1305 | 6.4050 ± 0.1315 | −0.2400 | 5.7690 | 5.5425 |
| 0 | Stirring | 5.8480 ± 0.0780 | 5.8330 ± 0.0980 | 5.9000 ± 0.1105 | −0.2570 | 1.1300 | 0.8755 |
| 5 | Stirring | 5.9565 ± 0.0330 | 5.9330 ± 0.0530 | 6.0500 ± 0.1000 | −0.3930 | 1.9285 | 1.5425 |
| 10 | Stirring | 5.5235 ± 0.0425 | 5.5165 ± 0.0750 | 5.5500 ± 0.1110 | −0.1270 | 0.6005 | 0.4745 |
| 0 | Spraying | 6.0410 ± 0.0250 | 5.9330 ± 0.0255 | 6.0500 ± 0.0985 | −1.8145 | 1.9285 | 0.1490 |
| 5 | Spraying | 6.0340 ± 0.0185 | 6.0330 ± 0.0205 | 6.0765 ± 0.0705 | −0.0165 | 0.7130 | 0.6965 |
| 10 | Spraying | 6.0355 ± 0.0190 | 6.0015 ± 0.0260 | 6.0535 ± 0.0980 | −0.5635 | 0.8560 | 0.2975 |
| | | Length, mm | | | | | |
| Control | | 30.6050 ± 4.0755 | 30.5375 ± 4.1250 | 30.9985 ± 4.2055 | −0.2210 | 1.4870 | 1.2695 |
| 0 | Stirring | 23.5265 ± 0.7755 | 23.4500 ± 0.7760 | 23.6330 ± 0.8805 | −0.3270 | 0.7755 | 0.4510 |
| 5 | Stirring | 28.6010 ± 0.9705 | 28.5830 ± 0.9905 | 28.6330 ± 0.8810 | −0.0610 | 0.1745 | 0.1135 |
| 10 | Stirring | 21.4850 ± 0.9100 | 21.4830 ± 0.9505 | 21.5165 ± 0.9715 | −0.0070 | 0.1550 | 0.1480 |
| 0 | Spraying | 31.5590 ± 0.1370 | 31.5500 ± 0.1415 | 31.6165 ± 0.2505 | −0.0290 | 0.2110 | 0.1820 |
| 5 | Spraying | 39.7200 ± 0.1365 | 39.4500 ± 0.1405 | 39.8330 ± 0.2510 | −0.6840 | 0.9620 | 0.2850 |
| 10 | Spraying | 29.5900 ± 0.1385 | 29.5480 ± 0.1415 | 29.6830 ± 0.3075 | −0.1415 | 0.4550 | 0.3140 |

The apparent density in pure pellets was 800.7505 kg·m$^{-3}$ immediately after pelleting, 793.0050 kg·m$^{-3}$ after freezing, and 728.8750 kg·m$^{-3}$ after defrosting (Table 3). Therefore, fluctuation of temperature slightly decreased the mass-to-volume ratio by 2.2835% in these products. The addition by spraying of liquid-state food waste boiled for 5 min caused the normal pellets to peak in apparent density at 1250.8500 kg·m$^{-3}$. Hence, very fine droplets certainly enabled the biomass particles to more efficiently diffuse into the voids, then more properly filling them and thus magnifying the density of the material. The mass-to-volume ratios of these products after freezing and defrosting were similar to the original at 1250.3000 and 1250.0665 kg·m$^{-3}$, respectively. Therefore, a reduction in apparent density in frozen and defrosted pellets with addition by spraying of liquid-state food waste boiled for the shortest amount of time statistically did not exist as per the values of 0.0440% and 0.0625%, respectively. The apparent density in pellets with addition by stirring was in the ranges of 805.0695–810.1785 kg·m$^{-3}$ immediately after pelleting, 800.0500–800.2300 kg·m$^{-3}$ after freezing, and 787.7165–795.3165 kg·m$^{-3}$ after defrosting, regardless of the time of boiling. Thus, thermal shock reduced the mass-to-volume ratio by 1.8685–2.2030% in these products, which were significantly softer than those with alternative addition by spraying. Therefore, adding liquid-state food by spraying by far was more efficient in both the formation of dense pellets and preservation of mass-to-volume ratio after freezing and defrosting than doing it conventionally by stirring.

The durability was 97.5045% for the pure pellets immediately after pelleting. This relatively high value substantially dropped to 91.0575% and 85.8510% in these products after freezing and defrosting, respectively (Table 4). The generation of fines from frozen and defrosted pellets with no addition proportionally increased by 7.0800% and 13.5740%, respectively, as a consequence of significant loss of capacity of the material to resist deformation by stress forces. Irrespective of the time of boiling, addition by spraying provided the normal, frozen, and defrosted pellets with meaningful durability in the ranges of 99.5165–99.7665%, 95.6665–99.2330%, and 95.2330–99.1500%, respectively. The addition by stirring, in turn, caused the pellets to be less durable after pelleting, freezing, and defrosting, with ranges of 99.3165–99.8000%, 93.8000–95.8330%, and 90.1000–93.8830%, respectively. Specifically, pellets with addition by stirring of liquid-state food waste boiled for 10 min were the most durable and, consequently, were less likely to release fines to the environment in normal storage conditions. However, these products themselves were not capable of firmly resisting thermal shock, as their durability dramatically declined by 6.3965% and 10.7660% after freezing and defrosting, respectively, while their content of fines increased by the same proportions. On the contrary, reduction

in durability in frozen and defrosted pellets with addition by spraying of liquid-state food waste boiled for 5 min practically did not exist, with values of 0.5380% and 0.6220%, respectively. Thereby, spraying was technically superior to stirring in preserving much of original durability after process of freezing–defrosting. This alternative method of introducing organic binder into the pelleting of residual biomass from distillation of cellulosic bioethanol spectacularly prevented both the frozen and defrosted pellets from releasing substantial quantities of particulates finer than 3.15 mm during their emptying from the container onto the working benchtop and during abrupt shifting from one place to another. The synergistic effects of binding the starting material with the liquid-state food waste on the non-inherent capacity of the pellet to resist to deformation by thermal shock became legibly more persistent with spraying than with stirring.

**Table 3.** Descriptive analysis for the apparent density of non-wood pellets with addition by stirring or spraying of liquid-state food waste subjected to the process of freezing–defrosting.

| Time | Method | Storage Condition | | | Fluctuation | | |
|---|---|---|---|---|---|---|---|
| | | Normal, N | Freezing, F | Defrosting, D | N–F | F–D | N–D |
| | | Apparent density, $kg \cdot m^{-3}$ | | | $\Delta$, % | | |
| Control | | 800.7505 ± 5.6750 | 793.0055 ± 5.9050 | 782.8750 ± 5.8875 | −0.9765 | −1.2940 | −2.2835 |
| 0 | Stirring | 805.0695 ± 1.2050 | 800.0500 ± 1.2350 | 787.7165 ± 1.2475 | −0.6275 | −1.5655 | −2.2030 |
| 5 | Stirring | 810.1785 ± 1.3755 | 800.2330 ± 1.4585 | 795.3165 ± 1.5035 | −1.2430 | −0.6180 | −1.8685 |
| 10 | Stirring | 807.2500 ± 1.3695 | 800.0665 ± 1.4090 | 790.5000 ± 1.2510 | −0.8980 | −1.2100 | −2.1190 |
| 0 | Spraying | 1055.2165 ± 1.0050 | 1050.5000 ± 1.0075 | 1050.2330 ± 1.0095 | −0.4490 | −0.0255 | −0.4745 |
| 5 | Spraying | 1250.8500 ± 1.1010 | 1250.3000 ± 1.1025 | 1250.0665 ± 1.1105 | −0.0440 | −0.0185 | −0.0625 |
| 10 | Spraying | 1051.3930 ± 1.0075 | 1050.6330 ± 1.0105 | 1050.0665 ± 1.0155 | −0.0720 | −0.0540 | −0.1260 |

**Table 4.** Descriptive analysis for the durability and generation of fines of non-wood pellets with addition by stirring or spraying of liquid-state food waste subjected to the process of freezing–defrosting.

| Time | Method | Storage Condition | | | Fluctuation | | |
|---|---|---|---|---|---|---|---|
| | | Normal, N | Freezing, F | Defrosting, D | N–F | F–D | N–D |
| | | Durability, % | | | $\Delta$, % | | |
| Control | | 97.5045 ± 1.0855 | 91.0575 ± 1.1005 | 85.8510 ± 1.1015 | −7.0800 | −6.0645 | −13.5740 |
| 0 | Stirring | 99.3170 ± 0.0055 | 94.6170 ± 0.0060 | 92.7000 ± 0.0045 | −4.9675 | −2.0675 | −7.1375 |
| 5 | Stirring | 99.5670 ± 0.0040 | 95.8330 ± 0.0075 | 93.8830 ± 0.0105 | −3.8955 | −2.0770 | −6.0535 |
| 10 | Stirring | 99.8000 ± 0.0075 | 93.8000 ± 0.0045 | 90.1000 ± 0.0275 | −6.3965 | −4.1065 | −10.7660 |
| 0 | Spraying | 99.5165 ± 0.0005 | 95.6665 ± 0.0010 | 95.2330 ± 0.0595 | −4.0245 | −0.4550 | −4.4975 |
| 5 | Spraying | 99.7665 ± 0.0010 | 99.2330 ± 0.0025 | 99.1500 ± 0.0020 | −0.5380 | −0.0835 | −0.6220 |
| 10 | Spraying | 99.7335 ± 0.0005 | 98.2000 ± 0.0030 | 97.6500 ± 0.0055 | −1.5615 | −0.5630 | −2.1335 |
| | | Fines, % | | | | | |
| Control | | 2.4955 ± 1.0855 | 8.9425 ± 1.1005 | 14.1490 ± 1.1015 | 7.0800 | 6.0645 | 13.5740 |
| 0 | Stirring | 0.6830 ± 0.0055 | 5.3830 ± 0.0060 | 7.3000 ± 0.0045 | 4.9675 | 2.0675 | 7.1380 |
| 5 | Stirring | 0.4330 ± 0.0040 | 4.1670 ± 0.0075 | 6.1170 ± 0.0105 | 3.8955 | 2.0770 | 6.0540 |
| 10 | Stirring | 0.2000 ± 0.0070 | 6.2000 ± 0.0045 | 9.9000 ± 0.0275 | 6.3965 | 4.1065 | 10.7660 |
| 0 | Spraying | 0.4835 ± 0.0005 | 4.3335 ± 0.0010 | 4.7670 ± 0.0595 | 4.0240 | 0.4550 | 4.4980 |
| 5 | Spraying | 0.2335 ± 0.0010 | 0.7670 ± 0.0025 | 0.8500 ± 0.0020 | 0.5375 | 0.0835 | 0.6220 |
| 10 | Spraying | 0.2665 ± 0.0005 | 1.8000 ± 0.0030 | 2.3500 ± 0.0055 | 1.5615 | 0.5630 | 2.1335 |

Normal, frozen, and defrosted pellets purely consisting of residual biomass started to break down 45.7500, 37.8950, and 33.7505 min after immersion in water, respectively (Table 5). Hence, the process of freezing–defrosting significantly reduced the capacity of the referential pellet to withstand deformation by water's abrasive forces by 35.553%. This was proof of the thermal shock imposing severe fractures over the structure of the material, thereby causing it to become more fragile and brittle and consequently to crumble more easily under the potential pressure of absorption of water from the

medium and atypical diffusion of it through breakpoints, cracks, and delamination on the aesthetically unpleasing surface. The addition by spraying of liquid-state food waste boiled for 5 min enabled the pellets to reach a peak in resistance to breaking down by water immediately after pelleting at 59.7665 min. Resistance to breaking down by water of these products neatly dropped to 54.6000 and 54.4500 min after freezing and defrosting, respectively. Evidently, the capacity of the pellet to resist deformation by water's abrasive forces increased quickly with spraying. However, irrespective of the time of boiling, thermal shock reduced the resistance to breaking down by water in frozen and defrosted pellets by 9.7640–13.2990%. Normal, frozen, and defrosted pellets with addition by stirring started to break down 57.5000–59.3330, 52.1000–54.3500, and 51.5000–54.1170 min after immersion, respectively, regardless of the time of boiling. The reduction in the resistance to breaking down by water of these products was in the range of 9.6390–12.930%. Therefore, stirring was technically comparable to spraying for this variable. Thus, mechanical instabilities by thermal shock could dramatically reduce the pellet's resistance to breaking down by water except when adding liquid-state food waste to the starting material, regardless of the method.

**Table 5.** Descriptive analysis for the resistance to breaking down by water of fines of non-wood pellets with addition by stirring or spraying of liquid-state food waste subjected to the process of freezing–defrosting.

| Time | Method | Storage Condition | | | Fluctuation | | |
|---|---|---|---|---|---|---|---|
| | | Normal, N | Freezing, F | Defrosting, D | N–F | F–D | N–D |
| | | Resistance to Breaking Down by Water, min | | | Δ, % | | |
| Control | | 45.7500 ± 5.9075 | 37.8950 ± 6.0050 | 33.7505 ± 5.9590 | −20.7285 | −10.9370 | −35.5535 |
| 0 | Stirring | 58.7665 ± 1.2045 | 52.5000 ± 1.0575 | 52.3330 ± 0.0085 | −11.9365 | −0.3175 | −12.2930 |
| 5 | Stirring | 59.3330 ± 1.0850 | 54.3500 ± 1.3025 | 54.1170 ± 0.0060 | −9.1690 | −0.4290 | −9.6390 |
| 10 | Stirring | 57.5000 ± 1.1260 | 52.1000 ± 1.2665 | 51.5000 ± 0.0095 | −10.3645 | −1.1515 | −11.1505 |
| 0 | Spraying | 58.8830 ± 0.4345 | 52.8665 ± 0.7440 | 52.4330 ± 0.0005 | −11.3810 | −0.8195 | −12.3015 |
| 5 | Spraying | 59.7665 ± 0.3360 | 54.6000 ± 0.7880 | 54.4500 ± 0.0005 | −9.4630 | −0.2745 | −9.7640 |
| 10 | Spraying | 58.5000 ± 0.4505 | 52.7500 ± 0.6575 | 51.6330 ± 0.0010 | −10.9005 | −2.1170 | −13.2990 |

The hygroscopicity was 14.5400%, 16.7050%, and 18.8565% for pellets with no addition after pelleting, freezing, and defrosting, respectively (Table 6). Therefore, thermal shock substantially extended the capacity of the pure pellet in a wet environment to absorb water from the atmosphere by 22.8380%, thus making it more hygroscopic. The hygroscopicity in normal, frozen, and defrosted pellets with addition by spraying was in the shortest ranges of 6.0215–8.1015%, 6.0570–8.1165%, and 6.0770–8.8415%, respectively, regardless of the time of boiling. Specifically, the hygroscopicity in pellets with addition by spraying of liquid-state food waste boiled for 5 min did not change significantly with freezing and defrosting, as shown by the values of 0.5890% and 0.9105%, respectively. Pellets with addition by spraying were much more hydrophobic than those with addition by stirring, for which the hygroscopicity was in the ranges of 10.3375–10.4685% immediately after pelleting, 10.4590–10.7000% after freezing, and 12.2685–12.9360% after defrosting. Thus, thermal shock increased the hygroscopicity in these products by 0.7910–2.2695% and 15.3615–19.1600%, respectively, regardless of the time of boiling and method of addition. Practically, spraying was the most efficient method of addition in both the production of hydrophobic pellets and preservation of their capacity to resist absorption of water naturally through the pores or even exceptionally through fractures on the surface finish after freezing and defrosting. Substantial improvements by very fine droplets in both the apparent density and polishing of the ends of structure certainly made these products morphologically and mechanically less prone to absorbing water from the environment.

**Table 6.** Descriptive analysis for the hygroscopicity of fines of non-wood pellets with addition by stirring or spraying of liquid-state food waste subjected to the process of freezing–defrosting.

| Time | Method | Storage Condition | | | Fluctuation | | |
|---|---|---|---|---|---|---|---|
| | | Normal, N | Freezing, F | Defrosting, D | N–F | F–D | N–D |
| | | Hygroscopicity, % | | | Δ, % | | |
| Control | | 14.5500 ± 1.3425 | 16.7050 ± 1.3385 | 18.8565 ± 1.5665 | 12.9003 | 12.8794 | 22.8380 |
| 0 | Stirring | 10.4685 ± 0.8750 | 10.5520 ± 0.9155 | 12.3685 ± 0.9045 | 0.7910 | 17.2145 | 15.3615 |
| 5 | Stirring | 10.3375 ± 0.9050 | 10.4590 ± 0.9070 | 12.7880 ± 0.9635 | 1.1615 | 22.2640 | 19.1600 |
| 10 | Stirring | 10.4575 ± 0.9705 | 10.7000 ± 0.9065 | 12.9360 ± 0.9580 | 2.2695 | 20.8920 | 19.1585 |
| 0 | Spraying | 8.1015 ± 0.05050 | 8.1165 ± 0.0510 | 8.8415 ± 0.0615 | 0.1850 | 8.9320 | 8.3695 |
| 5 | Spraying | 6.0215 ± 0.05000 | 6.0570 ± 0.0505 | 6.0770 ± 0.0580 | 0.5890 | 0.3245 | 0.9105 |
| 10 | Spraying | 7.6665 ± 0.05015 | 7.6860 ± 0.0670 | 7.8410 ± 0.0605 | 0.2495 | 2.0190 | 2.2235 |

The fractal dimensions were 1.9575, 1.9590, and 1.9595 for the mechanical fractures in referential pellets after pelleting, freezing, and defrosting, respectively (Table 7). Practically, breakpoints, cracks, and delamination showed the largest increases, with significant contraction and subsequent expansion by thermal shock in products with no addition. Irrespective of the time of boiling, addition by spraying caused the normal, frozen, and defrosted pellets to display the smallest fractures, as shown by the shortest fractal dimension ranges of 1.7490–1.7510, 1.7495–1.510, and 1.7495–1.7510, respectively. The spatial representation of fractures in pellets with addition by spraying of liquid-state food waste boiled for 5 min practically did not change with freezing and defrosting, as the thermal shock insignificantly increased the size of breakpoints and cracks by 0.0010–0.0140% and 0.0140–0.0285% respectively. The fractal dimension for fractures taking place on the pellets with addition by stirring was similar to that for those on the reference, regardless of the time of boiling and storage condition. The extension of damage by thermal shock was visually larger in pellets with addition by stirring. This was proof of the practicability and effectiveness of addition by spraying in generating stronger short-range and broad-range bonding mechanisms between and within biomass particles to assist pellets more resiliently in dealing with thermal shock by abrupt changes in the status of water from liquid to solid during freezing and then from solid to liquid during defrosting. This alternative method enabled the pellets to be more aesthetically pleasing, even after freezing and defrosting. On the contrary, the surface finish for pellets with addition by stirring was generally rougher, duller, and with more severe fractures (Figure S1, Supplementary Material).

**Table 7.** Descriptive analysis for the fractal dimension of fractures by thermal shock of non-wood pellets with addition by stirring or spraying of liquid-state food waste subjected to the process of freezing–defrosting.

| Time | Method | Storage Condition | | | Fluctuation | | |
|---|---|---|---|---|---|---|---|
| | | Normal, N | Freezing, F | Defrosting, D | N–F | F–D | N–D |
| | | Fractal Dimension of Fractures by Thermal Shock | | | Δ, % | | |
| Control | | 1.9575 ± 0.0355 | 1.9590 ± 0.0320 | 1.9595 ± 0.0340 | 0.0765 | 0.0255 | 0.1020 |
| 0 | Stirring | 1.9560 ± 0.0055 | 1.9570 ± 0.0055 | 1.9590 ± 0.0055 | 0.0340 | 0.1190 | 0.1530 |
| 5 | Stirring | 1.9555 ± 0.0040 | 1.9560 ± 0.0050 | 1.9580 ± 0.0055 | 0.0170 | 0.1280 | 0.1445 |
| 10 | Stirring | 1.9570 ± 0.0065 | 1.9575 ± 0.0060 | 1.9795 ± 0.0060 | 0.0170 | 1.1240 | 1.1280 |
| 0 | Spraying | 1.7505 ± 0.0045 | 1.7505 ± 0.0045 | 1.7510 ± 0.0045 | 0.0010 | 0.0280 | 0.0285 |
| 5 | Spraying | 1.7490 ± 0.0055 | 1.7495 ± 0.0055 | 1.7495 ± 0.0050 | 0.0095 | 0.0040 | 0.0140 |
| 10 | Spraying | 1.7510 ± 0.0040 | 1.7510 ± 0.0045 | 1.7510 ± 0.0045 | 0.0140 | 0.0030 | 0.0175 |

### 3.3. Interactive Effect of Time of Boiling, Method of Addition, and Storage Condition on the Physical–Mechanical Quality of Non-Wood Pellets

The range analysis did not fail to consistently predict the time of boiling and method of addition as the pair of factors with largest variability with respect to diameter, length, apparent density, and hygroscopicity, while the method of addition and storage of condition represented the pair of factors



with the largest variability relating to durability, fines, and fractal dimension of fractures by thermal shock. Time of boiling and storage condition comprised the pair of factors with the largest variability with respect to resistance to breaking down by water (Table S4, Supplementary Material). This method of applying descriptive analysis enabled the 2D fuzzy contour plotting approach to adequately determine the interactive effect of the sources of variation on the physical–mechanical quality of pellets.

### 3.3.1. Diameter and Length

The estimations for the effect of time of boiling on the diameter and length were negative, while the estimations for the effect of method of addition on these variables were positive (Table S5, Supplementary Material). The addition by spraying of liquid-state food waste boiled for 5 min should therefore yield the largest pellets transversally and longitudinally (Figure 2). In this condition, normal pellets were 6.0410 mm in diameter and 39.7200 mm in length. These values consistently decreased to 6.0355 and 29.5900 mm, respectively, with boiling of the supplemental ingredient for 10 min. Pellets with addition by stirring showed the shortest diameter and length. This was in line with the predictions at the relatively similar values of adjusted coefficient of determination ($R_{adj}^2$) of 0.6120 and 0.6000, respectively. Generally, both the factors contributed nearly equally to variability in these geometric features.

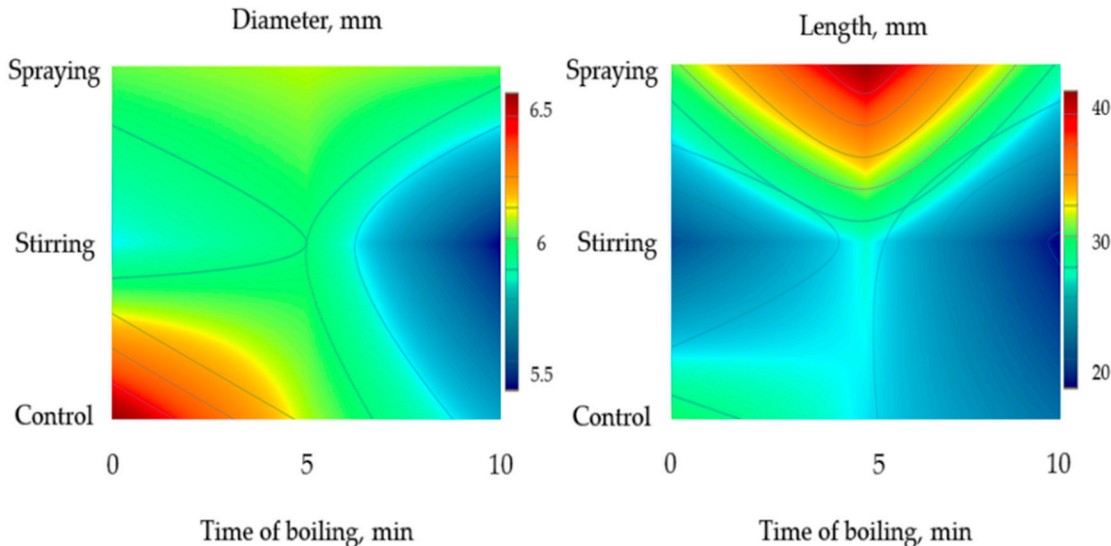

**Figure 2.** Two-dimensional fuzzy contour plotting for the interactive effect of time of boiling and method of addition on the diameter and length of non-wood pellets from pressing residual biomass from distillation of cellulosic bioethanol with liquid-state food waste.

### 3.3.2. Apparent Density

The estimation for the effect of time of boiling on the apparent density was negative but insignificant. On the contrary, the estimation for the effect of method of addition on this variable of physical–mechanical quality was positive and significant, regardless of the type of pellet. Pellets with addition by spraying impressively were 38.6050% denser than those with addition by stirring immediately after process of pelleting (Figure 3). This appreciable finding was consistent with the prediction at the relatively accurate value of $R_{adj}^2$ of 0.8450. Generally, the method of addition had a longer impact on this variable than time of boiling.

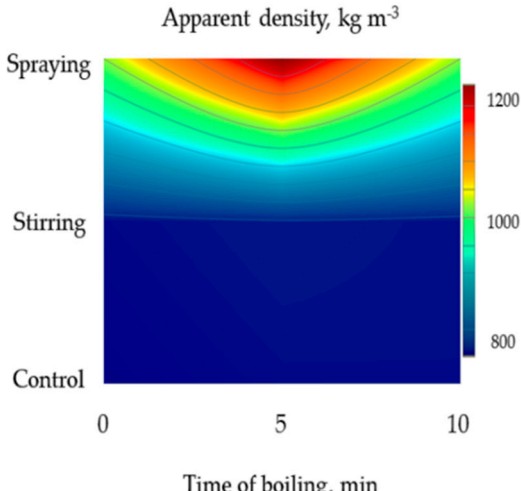

**Figure 3.** Two-dimensional fuzzy contour plotting for the interactive effect of time of boiling and method of addition on the apparent density of non-wood pellets from pressing residual biomass from distillation of cellulosic bioethanol with liquid-state food waste.

### 3.3.3. Durability and Fines

The estimation for the effect of method of addition on the durability was positive, while the estimation for the effect of the storage condition on this variable of mechanical stability was negative. Spraying should therefore enable residual biomass with liquid-state food waste to go smoothly through the machine to yield the most durable pellets immediately after compaction. Practically, normal pellets with addition by spraying showed increased durability (99.6720%); these products consistently reflected the most insignificant quantity of fines at 0.3280% (Figure 4). Irrespective of the method of addition, durability decreased by 4.5035% and 6.9135% with freezing and defrosting, while the generation of fines increased by the same proportions, respectively. These findings were in accordance with predictions at the relatively similar values of $R_{adj}^2$ of 0.7480 for durability and 0.7485 for the content of fines. Although accuracy of the response surface model for the durability was acceptable, the explanation for the highly complex capacity of the pellet to resist deformation by thermal shock would need to include factors other than those in this study in order to be precise and reliable. Generally, durability and fines both varied more greatly with respect to storage condition than with regard to the method of addition.

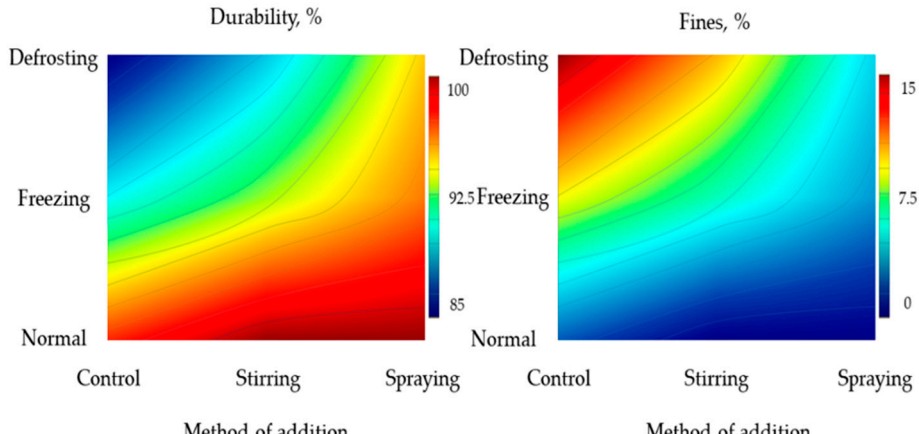

**Figure 4.** Two-dimensional fuzzy contour plotting for the interactive effect of method of addition and storage condition on the durability and fines of non-wood pellets from pressing residual biomass from distillation of cellulosic bioethanol with liquid-state food waste.

### 3.3.4. Resistance to Breaking Down by Water

The estimation for the effect of time of boiling on the resistance to breaking down by water was positive but insignificant. Normal pellets containing liquid-state material boiled for 5 min as part of their composition showed a peak resistance to breaking down by water of 59.5495 min (Figure 5). On the contrary, the estimation for the effect of storage condition on the resistance to breaking down by water was negative and significant. The capacity of pellet to sustain its structure for as long as possible against the pressure of absorption and diffusion of water should therefore decrease with freezing and defrosting, as the probabilities of the breakdown of bonds and occurrence of rough fractures through which water could enter the structure without any difficulty tended to become larger with thermal shock. Practically, frozen and defrosted pellets themselves started to break down 53.1945 and 52.7445 min after immersion in water, respectively. This trend endorsed the prediction at the value of $R_{adj}^2$ of 0.7125. Generally, the contribution of storage condition to variability for this variable was greater than that of time of boiling.

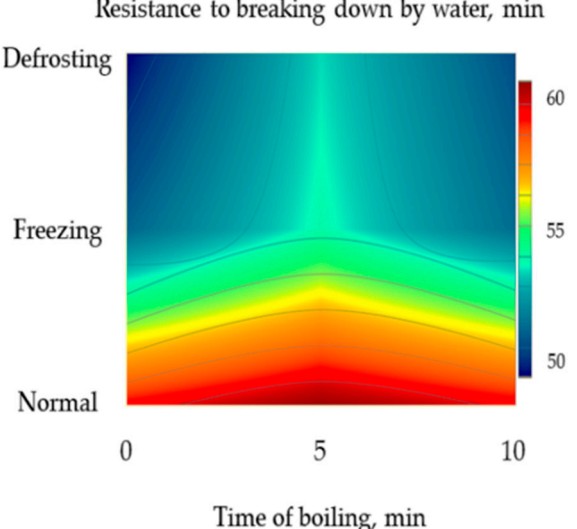

**Figure 5.** Two-dimensional fuzzy contour plotting for the interactive effect of time of boiling and storage condition on the resistance to breaking down by water of non-wood pellets from pressing residual biomass from distillation of cellulosic bioethanol with liquid-state food waste.

### 3.3.5. Hygroscopicity

The estimations for the effect of both the time of boiling and method of addition on the hygroscopicity were negative. However, this variable did not change significantly when maximizing the time of boiling. Normal pellets with addition by spraying showed increased hydrophobicity (Figure 6). These products were 29.8075% more hydrophobic than pellets with addition by stirring, consistent with the prediction at the value of $R_{adj}^2$ of 0.7725. Generally, this variable showed a higher range with respect to the method of addition than with time of boiling.

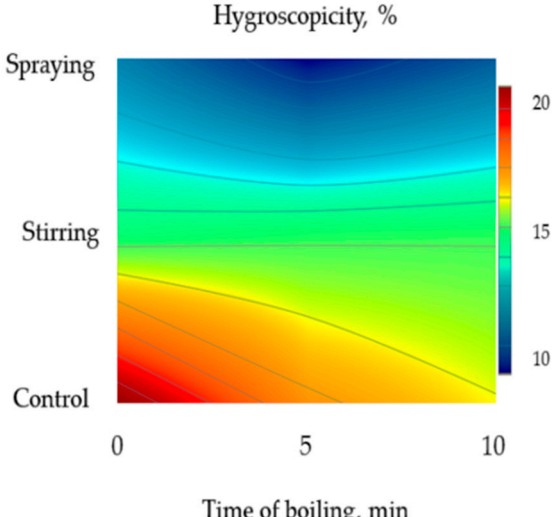

**Figure 6.** Two-dimensional fuzzy contour plotting for the interactive effect of time of boiling and method of addition on the hygroscopicity of non-wood pellets from pressing residual biomass from distillation of cellulosic bioethanol with liquid-state food waste.

### 3.3.6. Fractal Dimension of Mechanical Fractures by Thermal Shock

The estimation for the effect of method of addition on the fractal dimension of mechanical fractures was negative, while the estimation for the effect of storage on this non-Euclidean feature was positive. Breakpoints, cracks, splashing, and delamination in pellets with addition by spraying should therefore be reduced as compared to pellets with addition by stirring, whatever type of product, i.e., normal, frozen, and defrosted (Figure 7). The fractal dimensions of mechanical fractures in pellets with addition by spraying and stirring immediately after pelleting were 1.7505 and 1.9565, respectively. The thermal shock insignificantly magnified the spatial representation of mechanical fractures by 0.0080% and 0.0200% in frozen and defrosted pellets with addition by spraying, respectively. This commendable finding was in line with the prediction at the accurate value of $R_{adj}^2$ of 0.9905. Generally, the method of addition reflected a larger variability for the indicator of mechanical stability as comparison to storage condition.

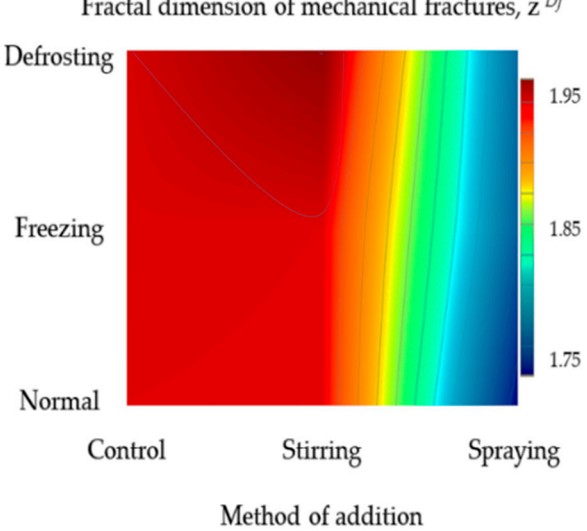

**Figure 7.** Two-dimensional fuzzy contour plotting for the interactive effect of method of addition and storage condition on the fractal dimension of fractures by thermal shock on non-wood pellets from pressing residual biomass from distillation of cellulosic bioethanol with liquid-state food waste.

### 3.3.7. Efficiency of Pelleting

The integration of boiling for the shortest amount of time with addition by spraying enabled the process of pelleting to increase the efficiency of conversion of biomass into pellets (95.5750%) (Figure 8). This value by far was larger than those found for the efficiency of compaction of material with addition by stirring (46.500–50.2500%), regardless of the time of boiling. Therefore, the operational performance of the machine increased quickly with spraying. The success of pelleting of biomass containing very fine droplets of liquid-state food waste as part of its composition supported the superiority of pellets with addition by spraying in terms of physical–mechanical behavior during freezing and defrosting.

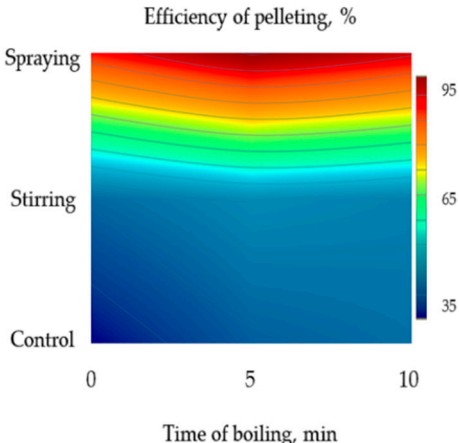

**Figure 8.** Two-dimensional fuzzy contour plotting for the interactive effect of time of boiling and method of addition on the efficiency of the process of pelleting in forming non-wood pellets from residual biomass from distillation of cellulosic bioethanol with liquid-state food waste.

### 3.4. Collinearities into the Addition of Liquid-State Food Waste to Make Freezing–Defrosting-Proof Pellets

The Pearson product–moment correlation test robustly tracked the most salient (multi)collinear patterns from the data set on the introduction of liquid-state food waste by spraying and stirring into the pelleting of residual biomass from distillation of cellulosic ethanol to produce freezing–defrosting-proof non-wood pellets (Figure S2, Supplementary Material). Density had positive linear relationships with both diameter ($r = 0.75$) and length ($r = 0.90$), but a negative linear relationship with the fractal dimension of mechanical fractures ($r = -0.95$). Geometrically larger pellets should therefore be denser, unless thermal shock causes them to undergo fractures that are fractally larger in dimension, thus simultaneously promoting loss of mass and porosity. Durability and hygroscopicity negatively correlated to each other ($r = -0.95$). Pellets with a lower mass-to-volume ratio should therefore absorb water from the atmosphere more easily than those higher in apparent density, as they should gain in availability of pores through the structure. Hygroscopicity unsurprisingly had positive correlation with the fractal dimension of fractures ($r = 0.70$). Therefore, the larger the size of the fracture, the greater the probability of the pellet of becoming more prone to absorbing water from the atmosphere through breakpoints, cracks, and delamination by thermal shock taking place on the surface and at the ends of the body. Durability had positive linear relationships with apparent density ($r = 0.55$) and resistance to breaking down by water ($r = 0.90$), but negative linear relationships with the hygroscopicity ($r = -0.60$) and, obviously, the fractal dimension of fractures ($r = -0.50$). More durable pellets should therefore be more hydrophobic and resist longer to crumbling due to the pressure of atypical absorption of water, as their fractures are less fractal in spatial representation. Thus, the larger the fractal dimension of the fracture, the greater the probability of the material releasing significant quantities of fines into the environment after pelleting, freezing, and defrosting. Globally, these collinearities endorsed the results of this study.

## 4. Discussion

*4.1. Relative Performance of Addition by Spraying of Liquid-State Food Waste to Produce
Freezing–Defrosting-Proof Non-Wood Pellets*

The range analysis, 2D fuzzy contour plotting approach, and fractal analysis by box-counting method collectively recognized and authenticated spraying as the most efficient method of adding liquid-state food waste to residual biomass from the distillation of cellulosic bioethanol to make pellets resistant to unacceptable but controllable roughness and instability as well as to significant loss of physical–mechanical quality due to thermal shock, while preserving the aesthetically pleasing visual aspect for the surface finish as much as possible after sudden changes in temperature during freezing at −20 °C and subsequent defrosting at room temperature. Besides markedly improving and maintaining the quality of the product, this alternative method enabled the process of pelleting to more efficiently convert the starting material into fuel pellets. This could be a result of noticeable improvements by very fine droplets in the homogeneity, grindability, flowability, compressibility, and compactability of the load of feeding. The granulometry of the material influences its behavior during pelleting [78]. The finer and more flexible the ingredient, the greater the probability of it more efficiently bonding particles closer together, thus optimizing compaction throughout [79]. This is of great relevance when dealing with development and implementation of cost-effective strategies to optimize the introduction and performance of additives in the compaction of biomass in order to manufacture handleable, transportable, and storable pellets for residential, commercial, and industrial applications requiring high-quality solid biofuels to go smoothly.

The explanation for the great performance of spraying in forming freezing–defrosting-proof pellets would be the spectrum of very fine droplets more uniformly distributing water-soluble binder on the layer of starting material before entering the pelletizer. This task of preproduction or preprocessing likely enabled the biomass particles to go smoothly through the channel-forming die underneath the pressing rollers of the machine during compaction to successfully bond together to shape themselves into more homogeneous and consistent pellets after thermosetting and cooling at room temperature. Thereby, pellets with addition by spraying became the densest, most durable, most hydrophobic, and were most resistant to severe mechanical fractures by thermal shock. Boiling for the shortest amount of time with addition by spraying certainly showed the most effective conditions for developing food waste as an anti-thermal shock binding agent to preserve the original physical–mechanical qualities as much as possible in pellets after the process of freezing and defrosting. This integration enabled the frozen and defrosted pellets to be technically comparable to normal pellets in terms of geometry, apparent density, durability and generation of fines, resistance to breaking down by water, hygroscopicity, and the fractal pattern of breakpoints, cracks, and delamination.

Pellets with addition by spraying ended up larger transversally and longitudinally. This may be a consequence of the very fine droplets more strongly and efficiently bonding more particles of biomass and liquid-state additive together, thus enabling them to form structures larger in both diameter and length. Augmentation in the geometric features of the pellet, depending on the scale, could make both the sizing and success of packing, transportation, and storage difficult and inefficient, and would create blockages in pneumatic systems requiring more compact products to perform without difficulties. The diameter and length both decreased in frozen pellets but increased in defrosted pellets. This presumable process of contraction and expansion was due to the status of water changing dynamically from liquid to solid during freezing and then from solid to liquid during defrosting. Crystallization of water at very low temperatures makes the distance between short-range bonding mechanisms shorter, thus shortening the pellet geometrically. The opposite of this physical phenomenon can effortlessly weaken the linkages by hydrogen bridges and Van der Waals forces between and within biomass particles, thus forcing the pellet to relax and become larger after defrosting [10,42].

Pellets with addition by spraying increased in apparent density immediately after pelleting. The very fine droplets of additive certainly more efficiently diffused into the voids and then more

properly filled them, thus reducing the availability of pores in the structure. Therefore, pellets with addition by spraying became significantly denser than those with addition by stirring. Therefore, the finer the additive and more homogeneous its distribution, the greater the probability of it successfully enabling layers of particles of biomass and liquid-state food waste to move, push, and press together during compaction to mold themselves into highly dense pellets. Beside improving density, addition by spraying more efficiently preserved much of original mass-to-volume ratio in frozen and defrosted pellets. This was proof of this alternative method, ensuring both the practicability and reliability of liquid-state food waste as a broad-range bonding force to assist pellets in more firmly resisting thermal shock by freezing and defrosting.

Pellets with addition by spraying increased in durability regardless of the condition of storage. If the initial durability of the pellet is high, its storage at freezing temperatures and its subsequent defrosting will not impact significantly on its capacity to resist deformation by shocking forces. The impressive efficiency of this method in both the formation of highly durable pellets after pelleting and preservation of much of original durability while reducing loss of fines to the environment after freezing and defrosting was likely due to strengthening by the organic binder of broad-range bonding mechanisms between and within biomass particles [18,19,80–83]. According to Zhang et al. [28], durability in pellets from pine sawdust with addition by spraying and stirring of sugary solution was 99.6000% and 99.5500%, respectively; yet spraying was technically superior to stirring in forming more mechanically stable structures, consistent with this study. Changes in temperature and relative humidity of the air during freezing at −19 °C and 90% and subsequent defrosting at 40 °C and 85% dramatically decreased the durability by up 4.3% in wood pellets [34]. In contrast, an insignificant reduction in durability was found in pellets with addition by spraying of liquid-state food waste boiled for the shortest amount of time after freezing and defrosting, at 0.6220%. This discrepancy endorsed the trend in this study with respect to the extent to which the alternative organic binding agent became an impressively powerful adhesive to make durable and consistent pellets upon freezing and defrosting. Very low temperatures significantly reduced the durability in pellets from chips, pinewood sawdust, sunflower husk, residues of corn, post-hydrolytic wood, and coal plus wheat straw, as pointed out by Dyjakon and Noszczyk [39]. The explanation by these authors for the reduction in durability was the destruction of bonds between biomass particles by water changing from liquid to solid during freezing and then from solid to liquid during defrosting. This transition usually causes the pellet to undergo severe mechanical fractures such as cracks, splashing, and delamination [34]. This was in line with the results of this study showing a significant decrease in durability and increase in fractal dimension of mechanical fractures in frozen and defrosted pellets purely consisting of residual biomass and with addition by stirring. The nature and composition of feedstock, pelleting operational conditions, properties of the pellet (e.g., water content, hardness and flexibility), and conditions of storage are factors determining the type, severity, and spatial representation of mechanical fractures [84]. Any expressive gain in storability or reduction in reactivity of the pellet to the storage can help to mitigate off-gas emissions [85]. This is of great relevance in drafting timely inferences about the potential of spraying liquid-state food waste to develop consistent pellets to assist dealing with this issue of storage.

The higher apparent density and more polished appearance on the outside of pellets with addition by spraying certainly enabled them to more firmly resist the absorption of water from the atmosphere through pores and even through eventual fractures by thermal shock [86,87]. These products were much more hydrophobic than those with addition by stirring. On the contrary, pellets with no addition or with addition by stirring had larger fractal patterns of breakpoints, cracks, and delamination in their ends and on their finish surface than those with addition by spraying, regardless of the time of pre-heating (graphical abstract). These products absorbed more water from the atmosphere, crumbled more easily, and lost their shape faster to water's abrasive forces. Thus, if the fracture is large in dimension, the pellet can no longer be mechanically consistent, as it loses in density, durability, and hydrophobicity, consistent with the outcomes of the Pearson product–moment correlation test.

Fractures in pellets with addition by spraying were the smallest in the fractal dimension. The spatial representation of these non-Euclidean patterns practically remained the same in frozen and defrosted pellets. The capacity of the pellet to absorb water from the atmosphere can vary drastically with changing temperature and relative humidity of the air during storage. If the pellet is highly hygroscopic, it cannot resist microbial degradation for long, thus rotting easily [88]. Other relevant negative aspects in the transport, storage, and processing of hygroscopic pellets include the real possibilities of fire and generation of pollutants [89,90]. Thus, pellets with addition by spraying of liquid-state food boiled for the shortest amount of time would be the most suitable pathways for the safe and efficient transport and storage of biomass, as their hygroscopicity was the lowest even after freezing and defrosting. This was the ultimate proof of the spraying standing out as the most effective method of addition of liquid-state food waste to make pellets mechanically resilient to thermal shock. This finding is appreciable when dealing with the development and implementation of strategies to transport and store pellets safely and efficiently in containers and indoor facilities with no heating systems, especially in the winter season.

### 4.2. Potential Applications for Non-Wood Pellets from Residual Biomass from Distillation of Cellulosic Bioethanol

Geometry is of great relevance when screening potential pellets for heating and power, as the length-to-diameter ratio affects ignition and combustion and influences the homogeneity of feeding, mainly in pneumatic systems [10,91,92]. Shorter pellets often make handling, transportation, and storage easier. They generally ignite faster but do not burn longer than larger pellets. Additionally, the shorter the pellet, the greater the improbability of it causing blockages in transport pipes in pneumatic systems [37]. Larger pellets, in turn, take longer to ignite but continue to burn for longer. However, they break more easily than shorter pellets under harsh conditions of handling, transportation, and storage, and require larger thermal conversion equipment or devices, entailing a larger investment cost for the customer. The ENPlus set the diameter and length to be in the ranges of 6–8 mm and 3.15–40 mm for the strictest class of residential biomass pellets, while the Initiative of Wood Pellet Buyers (IWPB) set the diameter to be in the ranges of 6–8 mm and the length to be no greater than 40 mm for the strictest class of industrial products (Table S6, Supplementary Material). Therefore, pellets with addition by spraying of liquid-state food waste boiled for the shortest amount of time would technically fall in the highest grids of both residential and commercial applications, as their diameter and length fulfill the requirements of international standards for these variables even after freezing and defrosting.

The apparent density is the measurement of mass-to-volume ratio of the pellet. This varies drastically with the nature and composition of the starting material, as well as with the conditions of pelleting, transportation, and storage [70,93,94]. The apparent density is pivotal to defining the size and success of packing, transportation, and storage. High apparent density usually indicates high-technical-quality products. Highly dense pellets often are more efficient and cheaper pathways to transporting and storing biomass in the long term than softer pellets. They generally release larger amounts of thermal energy during burning in thermal conversion equipment or devices [68,95]. Softer pellets, in turn, have lower energy density and require greater space for transportation and storage, making their logistics to some extent difficult, inefficient, and expensive. Another disadvantage of poorly dense pellets is the real possibility of misfeeding [29]. The ENPlus and IWPB set the apparent density to be at least 600 kg·m$^{-3}$, regardless of the application and class. Therefore, pellets with addition either by spraying or stirring of liquid-state food waste, boiled for the shortest amount of time, would be suitable for both residential and industrial applications, as their apparent density was in compliance with the requirement of international standards for this variable, even after freezing and defrosting. However, pellets with addition by spraying would persistently be much more attractive to potential consumers, as these are the densest, most stable, and most homogeneous products [45].

Durability is the measurement of the ability of the pellet to remain intact against deformation by multiple falls and collision forces. High durability usually implies high mechanical quality pellets [81,96]. Durable pellets resist longer to loss of biomass during emptying from the container onto the ground or during shifting them from one place to another. They show reliability in terms of transportation and storage, performance of thermal conversion equipment, and end user convenience [37]. Non-durable pellets, in turn, are more mechanically fragile and cannot support long-lasting stresses of handling, transportation, and storage such as fluctuating temperature and relative air humidity [34,97,98]. They release dust and fines more easily than durable pellets. Dust and fines reduce the fuel power of the pellet and cause wear of the thermal conversion equipment, thus increasing the need for maintenance and repair [42,99]. The ENPlus and IWPB set durability to at least 97.5% and the content of fines at no greater than 1% and 4% for the highest classes of residential and industrial products, respectively. Thus, pellets with addition by spraying of liquid-state food waste boiled for the shortest amount of time would fall in the strictest classes of residential and commercial products, as their durability and content of fines are in compliance with the guiding values of international standards for these covariables. These products would therefore be safe and efficient for transport and storage, and suitable for power biomass-burning household stoves, furnace-boiler systems for food, and coal-firing stations.

The quality assessment of pellets is certainly one of the most relevant tasks of post-production when setting critical values and accurate indicators, defining technical characteristics for thermal conversion equipment, informing the potential consumer about strengths and weaknesses of the product, ensuring legal compliance and security between stakeholders by defining responsibilities and duties, and disseminating environmentally friendly fuel. Length, diameter, durability, and content of fines assist in determining how suitable the pellet is for utilization in heat-and-power units. However, this analysis is complex and requires key variables other than those in this study to be more precise and reliable, such as heat of combustion and proximal and elemental contents of water, ash, and pollutants. International standards do not set guiding values for hygroscopicity and resistance to breaking down by water. However, the determination of these variables and fractal geometry of mechanical fractures would be timely and reliable in drafting inferences in order to understand and optimize the mechanical behavior of pellets to enable them to more efficiently address thermal stress by freezing and defrosting.

Globally, addition by spraying showed great technical potential in providing non-wood pellets from residual biomass from the separation/distillation of cellulosic bioethanol with mechanical attractiveness to power the strictest residential and industrial applications, even after freezing and defrosting. This alternative method enabled the normal, frozen, and defrosted pellets to be technically comparable to those from processing sugarcane bagasse [100–102], wheat straw [103], Moso bamboo [70], sewage sludge plus Chinese fir, camphor and rice straw [29], residues of olive tree [15], corn cob [104], and other woody and non-woody wastes [16,18,105–108] that are often found in the production of fuel pellets (Table S7, Supplementary Material).

## 5. Conclusions

The integration of short-term thermal-chemical preprocessing by boiling with addition by spraying rather than stirring can ensure both the practicability and reliability of liquid-state food waste as an effective anti-thermal shock organic binder to assist non-wood pellets in firmly withstanding unpredictable temperatures, which can decrease from 25 to −20 °C during freezing and then subsequently rise from −20 to 25 °C during defrosting. The discoveries of this comprehensive study focusing on waste management and biomass valorization are valuable for dealing with the development and implementation of cost-effective, environmentally-friendly downstream strategies aiming to improve physical–mechanical quality and preserve integrity to transport and store pellets safely and effectively in containers and indoor facilities while not systematically heating them in the winter season. The process of pelleting is an excellent waste-to-energy pathway to successfully reshape the organic fraction of MSW into an executable starchy broad-range mechanism to powerfully bond biomass particles together to form cohesive biosolids. This would be an option to assist in preventing

municipal administrations from expensively investing in the management of huge quantities of MSW, as well as the environment and society from experiencing adversities resulting from disposal in landfills or simply off-site irresponsibly and wastefully. Another appreciable aspect of this paper is the real possibility of converting residual biomass from separated/distilled cellulosic bioethanol to non-wood pellets with suitability for residential power and industrial applications requiring high technical quality products to perform. This would assist in managing this challenging co-product wisely to ensure second-generation bioethanol sustainable to fabricate and competitive with gasoline, while driving this technology towards the flex-factory concept.

**Supplementary Materials:** The following are available online at http://www.mdpi.com/1996-1073/13/12/3280/s1. Figure S1: Visual aspect of non-wood pellets from pressing residual biomass from distillation of cellulosic bioethanol with addition by spraying and stirring of liquid-state food waste, immediately after freezing. Figure S2: Correlogram for the linear relationships between variables of the physical–mechanical quality of non-wood pellets resulting from pressing residual biomass from distillation of cellulosic bioethanol with addition by stirring and spraying of liquid-state food waste subjected to the process of freezing–defrosting. Table S1: Technical specifications of integrated set of pelleting. Table S2: Norms for the physical–chemical characterization of the starting material and supplement. Table S3: Proximal and elemental properties of residual biomass from distillation of cellulosic bioethanol and food waste from restaurants. Table S4: Range analysis for the effect of time of boiling, method of addition, and storage condition on the variables of physical–mechanical quality and fractal dimension of mechanical fractures of non-wood pellets from residual biomass from distillation of cellulose bioethanol. Table S5: Analysis of lack-of-fit for the first-order surface response models for the interactive effect of time of boiling and method of addition on variables of physical–mechanical quality of non-wood pellets from pressing residual biomass from distillation of cellulosic bioethanol with liquid-state food waste. Table S6: Potential applications for non-wood pellets from processing residual biomass from distillation of cellulosic bioethanol with addition by stirring and spraying of liquid-state food waste boiled for five minutes. Table S7: Relative physical–mechanical quality of non-wood pellets from processing residual biomass from distillation of cellulosic bioethanol with addition by spraying of food waste boiled for five minutes.

**Author Contributions:** Conceptualization, B.R.d.A.M. and V.H.C.; Data curation, B.R.d.A.M. and C.T.M.; Formal analysis, B.R.d.A.M. and C.T.M.; Funding acquisition, R.d.S.V.; Investigation, V.H.C., P.A.M.d.F., and L.A.M.L.; Methodology, B.R.d.A.M., V.H.C., C.T.M., A.C.M. and J.C.C.; Project administration, R.d.S.V., P.A.M.d.F., L.A.M.L., and A.M.; Supervision, A.C.M. and J.C.C.; Writing—original draft, B.R.d.A.M. and V.H.C.; Writing—review and editing, P.R.M.L., S.B.R., and J.C.C. All authors have read and agreed to the published version of the manuscript.

**Funding:** This research was funded by São Paulo Research Foundation (FAPESP), grant number 24234-1.

**Conflicts of Interest:** The authors declare no conflict of interest.

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
