# Peer review of "Anti-Thermal Shock Binding of Liquid-State Food Waste to Non-Wood Pellets"

_energies, doi:10.3390/en13123280_

Round 1

Reviewer 1 Report

Comments regarding the manuscript entitled:

Anti-thermal shock binder of liquid-state food waste to non-wood pellets

The authors are presenting an interesting subject which is the exploration of food waste for the fabrication of non-wood pellets that can be used for heating.

Please correct:

Page 2 line 52 ‘off course’ by ‘of course’

‘The European Union (EU) is, off course, the world’s largest producer and consumer of wood pellets for heating and power’

Here the authors are presenting a statement that needs validation by introducing a reference.

Please correct this sentence:

‘this study is to purpose the production of’ with the first capital letter.

I will ask the authors to be consistent in the presentation of the introduction. As an example, Page 2 lines 49-63, the paragraph is about the production and consumption of pellets around the world and Europe then writing about the new technologies and going again to the production and consumption of pellets in Europe.

I find the introduction very long. I will suggest the authors make it shorter with a short paragraph linked to the world and economic aspect of the production of the pellets then the second paragraph will focus on technologies and new technologies used for the production of pellets and finally clearly state the aim of this study. On the same occasion, I ask the authors to go to ‘Energies’ there are new published studies going with the subject.

Is it possible to have a schema of the pilot-scale?

In section 2.1.2, what’s about the storage of the raw material before performing the experiment?

In section 2.2.2. Setting up, it is not clear if this was performed at the laboratory or somewhere else. If this was performed in the laboratory, in this case, we will need references for each step from drying to pelleting.

The authors did not mention the means used for the determination of all technical parameters with the errors.

Please correct:

Page 6 line 248

Data analysis

The authors must be more specific and give more details about the statistical analysis and provide is possible with the calculation of the error, standard deviation, etc. It seems a software was used. It this is the case, please provide more information about the software.

In general, we need to see and read more details about the different experimental setup.

Results

I find that the results can be more attractive with illustrations and table which we are missing in this manuscript and this make as ‘hard’ to follow the authors through the results. So, I suggest the addition of figures or tables.

Also, we are missing a comparison of the presented results we other published ones to see the trend or the improvement compared with a regular experiment or production.

I cannot understand the presented figures. I need to see more numbers nit just from 0 to 1, I find these representations not useful as they are presented.

And maybe put both figures together with and without to see the effect of the addition.

Reviewer 2 Report

This paper experimental investigated the physical and mechanical performances of a new binder for non-wood pellets. This topics is quite interesting and importance of this work is very high due to fact that this not only treated waste, but also have potential to reduce CO2. Also few work is made in such topic although it looks like not complex. In this paper, the better way to make presentation in the introductions shall be done. A reasonable flow of the introduction, either starting from waste, or starting introduction of gable warm is OK, but it shall be easy to be following. Introduction section shall be rewritten. Another issue is the title of this paper, a concise and straight link to the work shall be reflacted.

It is very interesting work.

minor comments are:

1, title: a more suitable title is needed

2, Introduction shall be reorganized with a logical flow, and make readers easy to follow  

Round 2

Reviewer 1 Report

The revised version of the manuscript is much clear than the first version. I have a couple of comments:

I will let the editor in chief decide about the title. My opinion is that the first one was more representative and maybe this is not the opinion of other reviewers.

 On page 3 line 120, I will suggest the following: The purpose of this study is the production...

I wanted to see references published by Energies and there is a lot of recent publications linked to the subject, that deserve to be cited. But this is just a suggestion. The authors have done enough literature review and enough work.
